# Vimentin intermediate filaments stabilize dynamic microtubules by direct interactions

Laura Schaedel[1,4], Charlotta Lorenz[1,4], Anna V. Schepers [1,2], Stefan Klumpp [2,3✉] & Sarah Köster [1,2✉]

The cytoskeleton determines cell mechanics and lies at the heart of important cellular functions. Growing evidence suggests that the manifold tasks of the cytoskeleton rely on the interactions between its filamentous components—actin filaments, intermediate filaments, and microtubules. However, the nature of these interactions and their impact on cytoskeletal dynamics are largely unknown. Here, we show in a reconstituted in vitro system that vimentin intermediate filaments stabilize microtubules against depolymerization and support microtubule rescue. To understand these stabilizing effects, we directly measure the interaction forces between individual microtubules and vimentin filaments. Combined with numerical simulations, our observations provide detailed insight into the physical nature of the interactions and how they affect microtubule dynamics. Thus, we describe an additional, direct mechanism by which cells establish the fundamental cross talk of cytoskeletal components alongside linker proteins. Moreover, we suggest a strategy to estimate the binding energy of tubulin dimers within the microtubule lattice.

[1] Institute for X-Ray Physics, University of Göttingen, Göttingen, Germany. [2] Max Planck School Matter to Life, Göttingen, Germany. [3] Institute for the Dynamics of Complex Systems, University of Göttingen, Göttingen, Germany. [4] These authors contributed equally: Laura Schaedel, Charlotta Lorenz. ✉email: stefan.klumpp@phys.uni-goettingen.de; sarah.koester@phys.uni-goettingen.de

The cytoskeleton is a dynamic biopolymer scaffold present in all eukaryotic cells. Its manifold tasks depend on the fine-tuned interplay between its three filamentous components: actin filaments, microtubules, and intermediate filaments (IFs)[1–6]. For example, all three types of cytoskeletal polymers participate in cell migration, adhesion, and division[3–6]. In particular, the interplay of IFs and microtubules makes an important contribution to cytoskeletal cross-talk, although the interaction mechanisms largely remain unclear[1,7–18].

For instance, vimentin, one of the most abundant members of the IF family, forms closely associated parallel arrays with microtubules in migrating cells[7,16,18]. Depolymerization of the microtubule network leads to a collapse of vimentin IFs to the perinuclear region, further attesting their interdependent organization in cells[9]. Several studies suggest that in cells, microtubules associated with the vimentin IF network are particularly stable: They exhibit increased resistance to drug-induced disassembly[9] and enhanced directional persistence during directed cell migration[18], and they are reinforced against lateral fluctuations[17]. Several proteins such as kinesin[8,11], dynein[13,15], plectin[1], and microtubule-actin cross-linking factor (MACF)[10,12] can mediate interactions between IFs and microtubules. These linker proteins may be involved in conferring microtubule stability to cells. However, the possibility that more fundamental, direct interactions independent of additional components like microtubule-associated proteins may contribute to the stability of microtubules remains unexplored. Such a mechanism could also explain the results of an in vitro study on dynamic microtubules embedded in actin networks: Depending on the network architecture, actin regulates microtubule dynamics and their lifetime. In particular, unbranched actin filaments seem to prevent microtubule catastrophe, thus stabilizing them, though the exact interaction mechanism is not revealed[19]. In contrast to the cell experiments that showed stabilization of microtubules by IFs, an earlier work found that many IFs, including vimentin, contain tubulin-binding sites and that short peptides containing these binding sites inhibit microtubule polymerization in vitro[14]. Yet, it is unknown how this effect relates to fully assembled vimentin filaments. Indeed, studying such reconstituted in vitro systems provide essential information for the understanding of hybrid biopolymer materials, including their rheological properties and polymerization kinetics.

Here, we studied these interactions by combining in vitro observations of dynamic microtubules in the presence of vimentin IFs with single-filament interaction measurements and complementary numerical simulations. In stark contrast to ref. 14, our observations and simulations of dynamic microtubules reveal a stabilizing effect by the surrounding vimentin IFs. Based on our experimental data, we also estimated the tubulin dimer binding energy within the microtubule lattice, which is a much sought-after parameter for understanding microtubule dynamic instability[20–27]. This value has previously only been determined by molecular dynamics simulations and kinetic modeling[20,22,27] or by using atomic force microscopy to indent stabilized microtubules[24].

## Results

**Dynamic microtubules in the presence of vimentin**. To study the influence of IFs on microtubule dynamics, we polymerized microtubules in the presence of vimentin IFs. We imaged the microtubules by total internal reflection fluorescence (TIRF) microscopy as sketched in Fig. 1a. As nucleation sites for dynamic microtubules, we used GMPCPP-stabilized microtubule seeds (green in Fig. 1a) adhered to a passivated glass surface. For simultaneous assembly of microtubules (cyan) and IFs (red), we

supplemented a combined buffer (CB) containing all ingredients necessary for the assembly of both filament types with 20 or 25 μM tubulin dimers and 2.3 or 3.6 μM vimentin tetramers (0.5 or 0.8 g/L protein). All experiments presented in this work refer to these protein concentrations. All TIRF experiments were carried out in the same buffer conditions. Figure 1b shows a typical fluorescence image of mixed microtubules and vimentin IFs.

We analyzed the microtubule dynamics using kymographs obtained from TIRF microscopy as shown in Fig. 1c. As expected[28], the microtubule growth rate increased at the higher tubulin concentration (Fig. 1d, cyan). Yet, the presence of vimentin IFs did not affect the growth and depolymerization rates: on average, the differences between the medians of the different conditions correspond to <3% and <5% of the data range for the growth and depolymerization rates, respectively (Fig. 1d, e). Interestingly, we observed a marked decrease in the catastrophe frequency[29] in the presence of vimentin IFs at both tubulin concentrations (Fig. 1f, red and cyan stripes). Moreover, vimentin IFs promote microtubule rescue (Fig. 1g). As rescue events are rare at the lower tubulin concentration[29], we only report the rescue data for 25 μM tubulin. When assembly is initiated, vimentin unit-length filaments form after about 100 ms.[30] Therefore, we assume that vimentin filaments, not precursors, interacted with the microtubules. In addition, we did not observe differences in microtubule dynamics when comparing early and late time points within the same experiment, although the mean vimentin filament length increased over the course of the experiments (see Supplementary Fig. 1). These results indicate that vimentin IFs stabilize dynamic microtubules by suppressing catastrophe and enhancing rescue, while leaving the growth rate unaffected. A higher vimentin concentration enhances these effects.

**Interaction forces between microtubules and vimentin filaments**. From these observations, we hypothesize that there are direct, attractive interactions between microtubules and vimentin IFs that stabilize dynamic microtubules. To test this hypothesis, we studied the interactions of single stabilized microtubules and vimentin IFs using optical trapping (OT), a complementary method to our TIRF experiments, as illustrated in Fig. 2. We prepared fluorescent and biotin-labeled microtubules and vimentin IFs as sketched in Supplementary Fig. 2. We used an OT setup combined with a microfluidic device and a confocal microscope (LUMICKS, Amsterdam, The Netherlands) to attach a microtubule and a vimentin IF to separate bead pairs via biotin-streptavidin bonds as shown in Fig. 2a and Supplementary Fig. 3a. Once the IF and microtubule were in contact, we moved the IF back and forth in the $y$-direction. If the IF and microtubule interacted, eventually either the IF-microtubule interaction broke (Fig. 2b) or the IF-microtubule interaction was so strong that the microtubule broke off a bead (Fig. 2c). To study the orientation dependence of the interaction, we included two additional measurement geometries: (i) we turned the microtubule by 45° as shown in Fig. 2e or (ii) moved the IF horizontally in the $x$-direction along the microtubule (Fig. 2f). We categorized the type of interaction, i.e., no interaction, the IF-microtubule bond broke, or the microtubule broke off the bead, for each filament pair, as shown by pictograms in Fig. 3a, top.

With the OTs, we recorded the force $F_{1y}$ or $F_{1x}$ that acted on trap 1 (see Fig. 2d and Supplementary Fig. 3b), which increased after the IF bound to the microtubule. Based on the geometry of the filament configuration from the confocal images, we calculated the total force $F_C$ that the IF exerted on the microtubule (see Supplementary Fig. 3c). In Fig. 3b we show

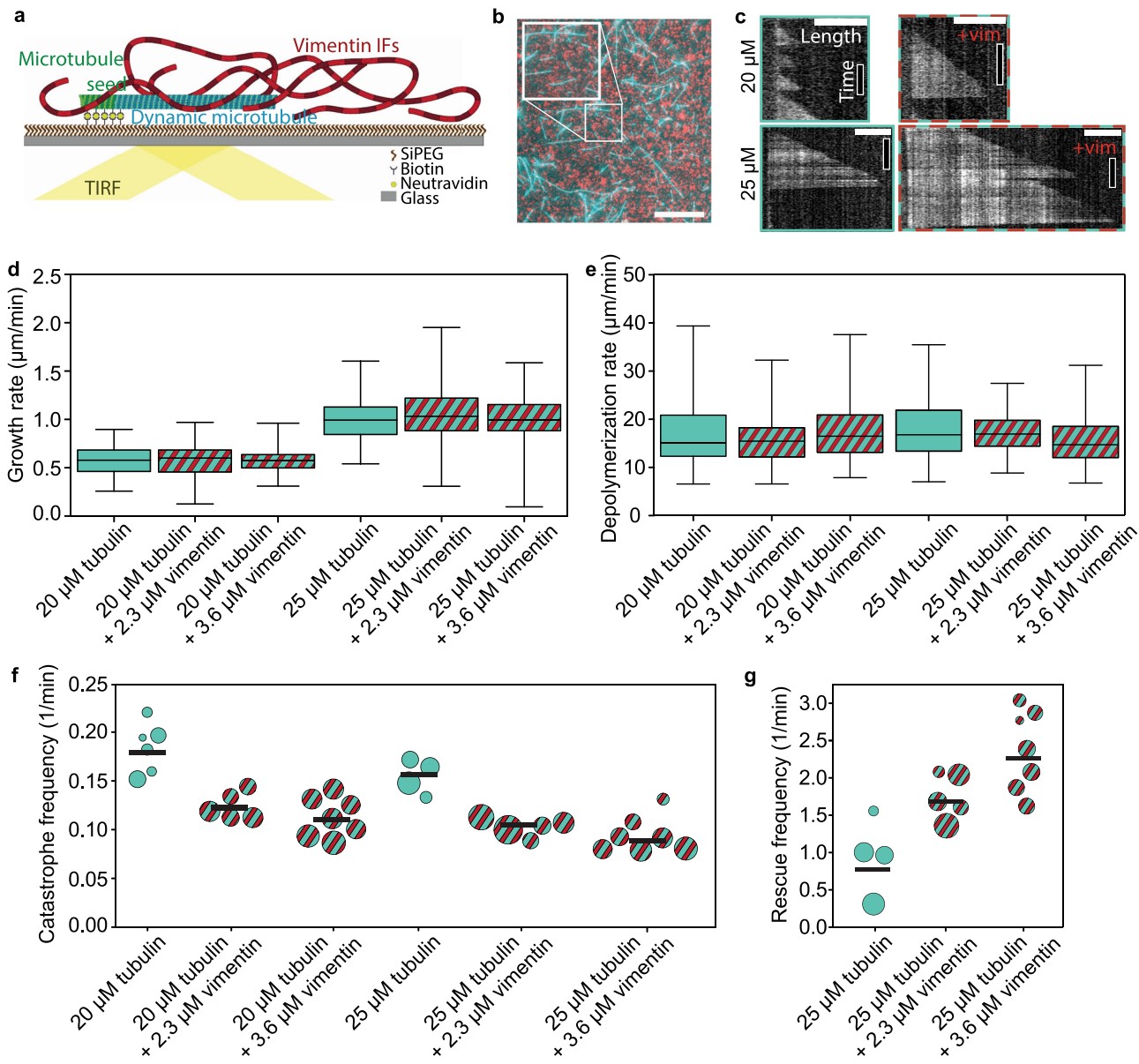

**Fig. 1 Vimentin IFs stabilize dynamic microtubules. a** Illustration of the experimental setup. We attached microtubule seeds (green) to a biotin-polyethylene glycol-silane (biotin-SiPEG) coated cover glass. Dynamic microtubules (cyan) grew from the seeds. Vimentin IFs (red) formed an entangled, fluctuating network. We imaged microtubules by TIRF microscopy. **b** Fluorescence micrograph of microtubules (cyan) embedded in a vimentin IF network (red). The inset shows the enlarged detail. Scale bar: 10 μm. **c** Example kymographs of microtubules growing at 20 or 25 μM tubulin in the presence (+vim; 2.3 μM) or absence of vimentin. Scale bars: 3 μm and 5 min. **d**, **e** Vimentin does not affect the microtubule growth and depolymerization rates, irrespective of the tubulin concentration. Cyan boxplots represent experiments with tubulin only; cyan- and red-striped boxplots illustrate experiments with tubulin and vimentin. Boxplots include the median as the center line, the 25th and 75th percentiles as box limits, and the entire data range as whiskers. In (**d**), for 20 μM tubulin and in the absence of vimentin, the boxplot represents the data from $N = 6$ samples and $n = 106$ growth events (we define a growth event as the new outgrowth of a microtubule from the seed or after a rescue event). For the other boxplots, from left to right, $N = 5, 7, 4, 5$, and $7$, respectively, and $n = 163, 282, 113, 160$, and $218$, respectively. In (**e**), from left to right, the boxplots represent the data from $n = 58, 140, 228, 112, 133$, and 165 depolymerization events, respectively. **f** The catastrophe frequency of the microtubules decreases in the presence of vimentin. Each circle represents an experiment including multiple microtubules. The area of the circle scales with the total summed microtubule growth time of the respective experiment. Black bars indicate the weighted mean. From left to right, the plot represents the data from a total of 805, 1469, 2692, 1142, 2203, and 2232 min of growth time, respectively. **g** Vimentin enhances the microtubule rescue frequency. Each circle represents an experiment including multiple microtubules. The area of the circle scales with the total microtubule depolymerization time. From left to right, the plot represents the data from a total of 51, 71, and 52 min depolymerization time, respectively. All tubulin and vimentin concentrations are input concentrations. Source data are provided as a Source Data file.

the resulting force calculated for the data shown in Fig. 2d. Combining all experiments with a breaking IF-microtubule bond leads to a distribution of $n_i$ breaking forces $F_B$ as shown in the force histogram in Fig. 3c. Due to thermal fluctuations, the force resolution of our system is limited to 1 pN and we thus focused on interaction forces above 1 pN, which is consistent with physiologically occurring intracellular forces. The breaking forces are in the range of 1–65 pN, with higher forces occurring less often. Hence, in agreement with our hypothesis, our experiments show that single microtubules and vimentin IFs directly interact,

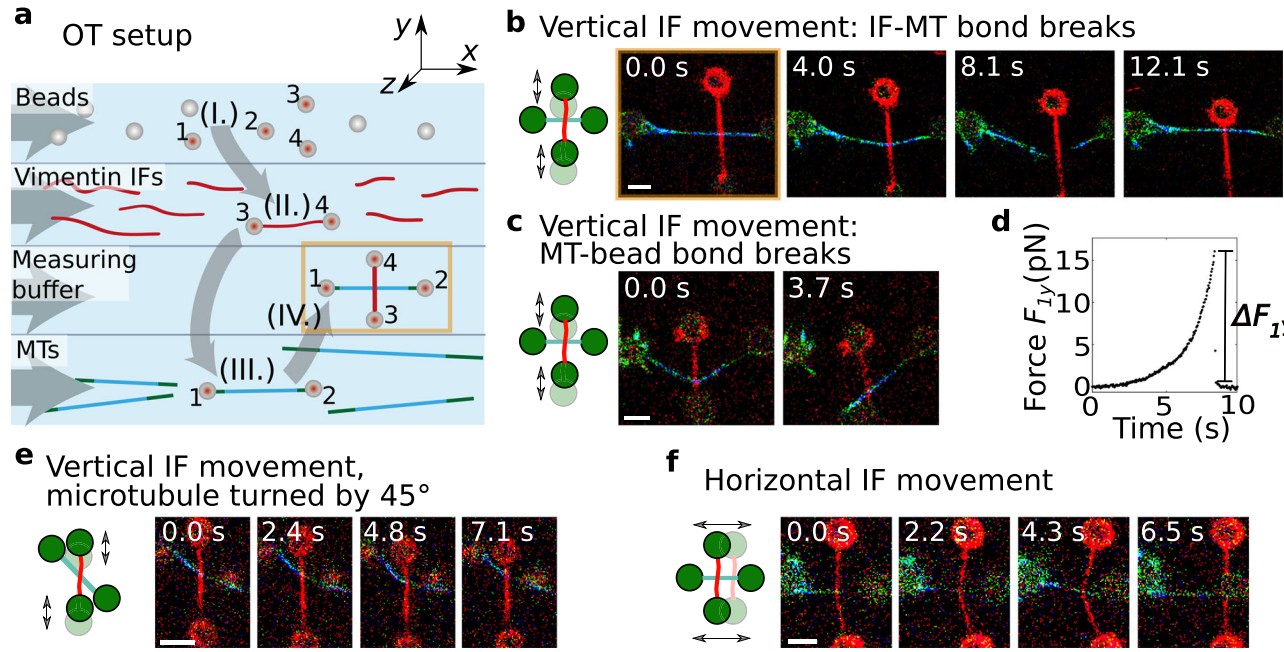

**Fig. 2 Direct interactions between stabilized microtubules and vimentin IFs. a** Schematic of the setup for the OT experiments for interaction measurements in microfluidic flow channels. Four streptavidin-coated beads were captured by OTs (I.). We used one bead pair (beads 3 and 4) to attach a vimentin IF (II., red), and the other bead pair (beads 1 and 2) to attach a microtubule (III., green-cyan). We brought the IF and the microtubule into contact in a crossed configuration (IV.). Next, we moved the IF perpendicularly to the microtubule to study the IF-microtubule interactions while we took confocal fluorescence images (starting position marked in yellow in (**a**) and (**b**)). **b** Typical confocal fluorescence images of an IF-microtubule interaction which broke while the IF was moved vertically and **c** a strong IF-microtubule interaction for which the microtubule broke off the bead. **d** Typical experimental force increase $F_{1y}$ on bead 1 in the $y$-direction once a bond forms. Breaking of the force causes a force jump of $\Delta F_{1y}$. **e** Typical confocal fluorescence images of a breaking IF-microtubule interaction at a 45° angle between them while the IF was moved vertically. **f** Typical confocal fluorescence images of a breaking IF-microtubule interaction in perpendicular configuration while the IF was moved horizontally. Scale bars: 5 µm. Measurements were conducted on at least three different days. Source data are provided as a Source Data file.

i.e. without involving any linker proteins, and that these interactions can become so strong that forces up to 65 pN are needed to break the bonds. This range of forces is physiologically relevant and comparable to other microtubule-associated processes: Single microtubules can generate pushing forces of 3–4 pN while forces associated with depolymerization can reach 30–65 pN[31]. Kinesin motors have stalling forces on the order of a few pN[32].

To better understand the nature of the interactions between single microtubules and vimentin IFs, we varied the buffer conditions in which we measured the filament interactions. First, we probed possible hydrophobic contributions to the interactions by adding 0.1% (w/v) Triton-X 100 (TX100), a non-ionic detergent. Rheological studies of IF networks previously suggested that TX100 inhibits hydrophobic interactions[33]. Tubulin dimers have several hydrophobic regions as well[34], some of which are accessible in the assembled state[35]. As shown in Fig. 3d and e, the number of interactions decreases and the breaking forces are slightly lower in the presence of TX100 than in pure CB. We calculated the binding rate $r_{b,\text{eff}}$ by dividing the total number of interactions larger than 1 pN by the time for which the two filaments were unbound: TX100 leads to a lower binding rate $r_{b,\text{eff,TX100}} = 0.56 \times 10^{-2}\,\text{s}^{-1}$ compared to the binding rate $r_{b,\text{eff},y} = 1.1 \times 10^{-2}\,\text{s}^{-1}$ without TX100. We speculate that TX100 interferes with the binding sites on both filament types by occupying hydrophobic residues on the surface of the filaments and thereby inhibits hydrophobic interactions between the biopolymers[33]. Consequently, the reduced number of interactions in the presence of TX100 indicates that hydrophobic effects contribute to the interactions.

Next, we tested for electrostatic contributions to the interactions by adding magnesium chloride to the buffer. When probing

interactions in CB buffer with a total concentration of 20 mM magnesium, we observed both an increase in strong interactions, where the IF pulls the microtubule off a bead, and higher breaking forces (Fig. 3aIII. vs. fIII. and c vs. g). The binding rate of microtubules and vimentin IFs increases to $r_{b,\text{eff,Mg}} = 1.3 \times 10^{-2}\,\text{s}^{-1}$. Generally, charged, suspended biopolymers in the presence of oppositely charged multivalent ions have been shown to attract these ions, leading to counterion condensation along the biopolymers. Consequently, the filaments attract each other through overscreening[36,37]. Our data are in agreement with this effect. At high magnesium concentrations, bonds are more likely to form and become stronger. Note that for both added magnesium chloride and TX100, the intermediate interactions (II) are decreased compared to the control conditions, due to a higher percentage of strong interactions (III) or weak interactions (I), respectively. Therefore, we conclude that both hydrophobic and electrostatic effects contribute to the direct interactions between microtubules and vimentin IFs.

When we moved the IF across the microtubule at an angle of 45° or horizontally in the direction of the microtubule (see Figs. 2e, f, 3h–k), we observed an increased binding rate ($r_{b,\text{eff},45°} = 1.6 \times 10^{-2}\,\text{s}^{-1}$ and $r_{b,\text{eff},x} = 2.4 \times 10^{-2}\,\text{s}^{-1}$, respectively). This increase can be explained by an increased encounter rate of potential binding sites due to the different geometries (see Supplementary Information). Taking into account this geometric factor, we calculated the probability of a microtubule binding to a vimentin IF $p_{\text{IF-MT}}$ for the different geometries and obtained $p_{\text{IF-MT}} \simeq 6.1 \times 10^{-4}$ per pair of vimentin unit-length filament and tubulin dimer, independent of the geometry. The breaking forces were found to be similar for the three different geometries (Fig. 3c, i and k). To test if vimentin IFs and microtubules

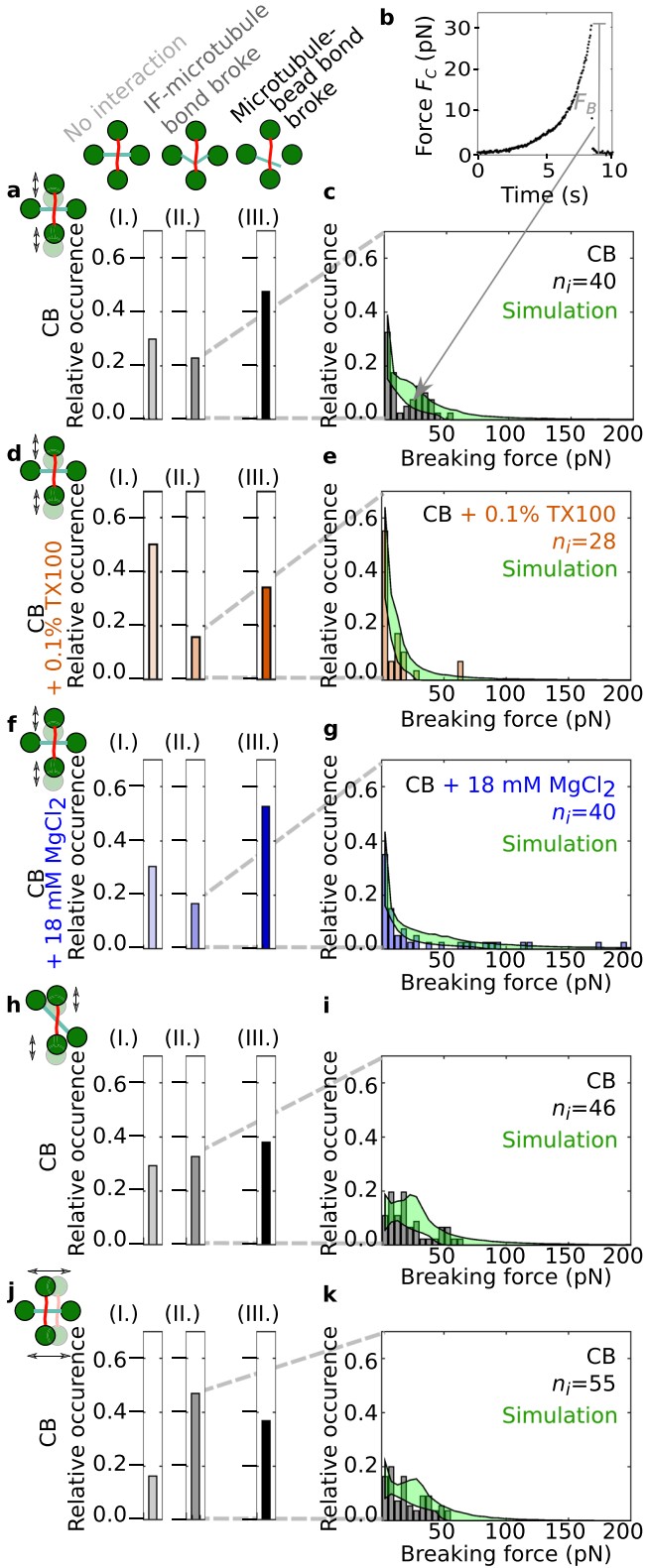

**Fig. 3 Hydrophobicity and electrostatics contribute to the IF-microtubule interactions. a** We classified the interactions between IF-microtubule pairs into three different groups as shown by the pictograms: no interaction (I.), breaking of the IF-microtubule bond as shown in Fig. 2b, e, and f (II.), and breaking of the microtubule-bead bond as shown in Fig. 2c (III.). **b** Typical experimental force-time behavior of the IF-microtubule bond showing the total force acting on the IF-microtubule bond, $F_C$. The plot represents the corrected version of the force data shown in Fig. 2d taking into account the geometry of the filament configuration. **c** Histograms of $n_i$ experimentally recorded breaking forces (gray) and simulated data (green) for the measurements in pure CB when the IF-microtubule interaction broke as shown in Fig. 2b. Due to statistical fluctuations, the distribution appears to be bimodal; however, it can still be well described with a unimodal distribution. **d** TX100 (orange) suppresses some of the interactions, which results in more IF-microtubule pairs without any interaction (**a**I. vs. **d**I.) and fewer instances of IF-microtubule interactions (**a**II. and III. vs. **d**II. and III.). **e** The IF-microtubule bonds formed in the presence of TX100 break at lower forces. **f** Magnesium (blue) does not change the relative number of IF-microtubule pairs that do not interact (**a**I. vs. **f**I.), but leads to fewer IF-microtubule breaking events (**a**II. vs. **f**II.) because the interactions become so strong that the microtubule breaks off the bead more often (**a**III. vs. **f**III.). **g** The IF-microtubule bonds formed in the presence of additional magnesium break at higher forces. **h** When the microtubule was turned by 45°, IF and microtubule interacted more frequently (**h**II. vs. **a**II.). **i** The corresponding distribution of the breaking forces resembles the distribution for the perpendicular geometry (see **c**). **j, k** The binding rate for a horizontal movement of the IF increases compared to a vertical movement (**j**II. vs. **a**II.). **k** The corresponding distribution of breaking forces is similar to the distributions for the other two geometries (see **i** and **c**). Source data are provided as a Source Data file.

accessible experimentally, we applied a modeling approach. Due to the experimentally observed independence of the measuring geometry, we applied a one-dimensional transition model. It should be noted, however, that for experimental systems with a geometry dependence, the model would have to be replaced by a more complex model. We modeled the IF-microtubule interaction as a single molecular bond with force-dependent stochastic transitions between the bound and unbound state. The time-dependent force increase $F(t)$ has an entropic stretching contribution[38,39] for forces below 5 pN and increases linearly for higher forces as observed in the experiment[40,41]. We assume that the binding (b) and unbinding (u) rates $r_b$ and $r_u$, respectively, depend on the applied force, an activation energy $E_{Ab}$ or $E_{Au}$, the thermal energy $k_BT$, and a distance $x_b$ or $x_u$ to the transition state, which is on the order of the distance between the IF and the microtubule at the site of the bond:

$$r_b(t) = r_{b,0} \exp\left(\frac{-E_{Ab}}{k_BT}\right) \cdot \exp\left(\frac{-F(t)x_b}{k_BT}\right) ,$$
$$r_u(t) = r_{u,0} \exp\left(\frac{-E_{Au}}{k_BT}\right) \cdot \exp\left(\frac{F(t)x_u}{k_BT}\right) .$$
$$(1)$$

We summarize the force-independent factor in Eq. (1) as an effective zero-force rate:

$$r_{b,\text{eff}/u,\text{eff}} = r_{b,0/u,0} \exp\left(\frac{-E_{Ab/Au}}{k_BT}\right) . \qquad (2)$$

In contrast to the force and the effective binding rate $r_{b,\text{eff}}$, neither $r_{u,\text{eff}}$ nor $x_b$ or $x_u$ can be determined from our experimental data. Due to detailed balance, the sum $x_b + x_u$ is constant[42]. Since we only observed a small number of rebinding events under force, we focused on the unbinding processes and studied $x_u$. Hence, we simulated IF-microtubule interactions for

co-align due to their interaction, as reported for migrating cells[18], we relaxed the vimentin IF in the optical trap to allow for "zipping" events, or mixed the filaments in solution, but did not observe spontaneous bundling.

**Two-state model of the interactions.** For a more profound understanding of the physical bond parameters, which are not

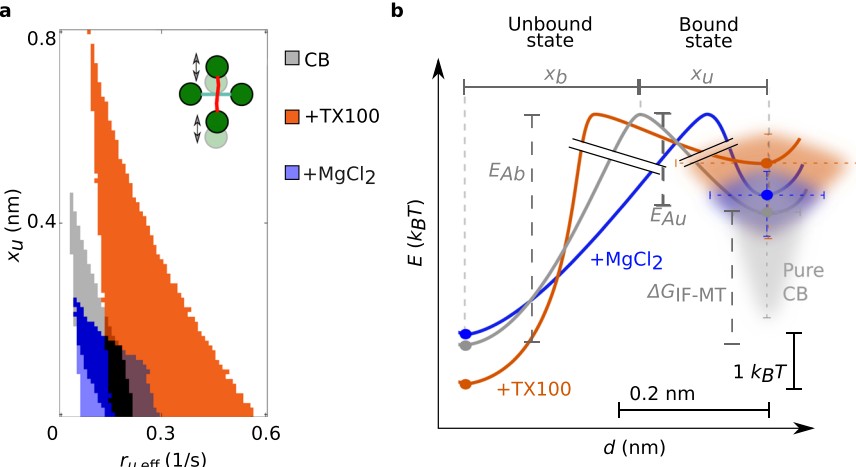

**Fig. 4 Hydrophobic and ionic reagents change the IF-microtubule bond properties. a** Valid unbinding rates $r_{u,\text{eff}}$ and potential widths $x_u$ to simulate the experimental data shown in Fig. 3c, e, and g for the different buffer conditions: pure CB (gray), CB with TX100 (orange), and CB with additional magnesium (blue). $r_{u,\text{eff}}$ and $x_u$ pairs, which are valid for several buffer conditions, are color coded by mixed colors. $r_{u,\text{eff}}$ and $x_u$ increase from additional magnesium chloride across pure CB to added TX100. **b** Energy landscape for the theoretical modeling of the IF-microtubule bond: A two-state model (unbound, bound) is sufficient to describe the experimental data shown in Fig. 3. From the binding and unbinding rates, we calculated the differences in activation energies $E_{\text{Ab}}$ and $E_{\text{Au}}$ (see Eq. (8) in the Supplementary Information) of bonds in different buffer conditions to open or close. However, the absolute values cannot be determined, as indicated by the graph break (black double-lines). For different buffer conditions, the position of the energy barrier relative to the unbound and bound state, $x_b$ and $x_u$, respectively, changes. Source data are provided as a Source Data file.

different sets of $r_{u,\text{eff}}$ and $x_u$ and compared the resulting distributions of breaking forces to our experimental data. We accepted the tested parameter sets if the distributions passed the Kolmogorov-Smirnov test with a significance level of 5%. The minimum and maximum of all accepted simulation results, shown as the borders of the green areas in Fig. 3c, e, g, i, and k, agree well with the experiments. Figure 4a shows all accepted parameter pairs $r_{u,\text{eff}}$ and $x_u$ for the different buffer conditions (color code: gray (pure CB), orange (CB with TX100), blue (CB with additional magnesium); corresponding mixed colors for regions, where valid parameters overlap). Both parameters increase from additional magnesium (blue) across no addition (gray) to added TX100 (orange). A corresponding diagram for comparison of the different measuring geometries is shown in Supplementary Fig. 4. Whereas the force-free factor of the unbinding rate does not depend on the geometric configuration, the force-dependent factor is slightly more sensitive to force for a horizontal movement of the IF or a vertical movement of the IF, with the microtubule turned by 45° than for a vertical movement of the IF perpendicular to the microtubule. To understand these data more intuitively, we calculated the energy diagrams, as plotted in Fig. 4b, using Eq. (8) (see Supplementary Information) considering the same buffer condition in unbound and bound (1 and 2) state or different buffer conditions (1 and 2) and the same state.

Surprisingly, both TX100 and additional magnesium only mildly affect the activation energies. Yet, for TX100 we observed a marked increase in distance to the transition state, $x_u$ (compare Fig. 4b orange to gray), which we interpret as a "looser binding" between the IF and the microtubule. Thus, the force-dependent term in Eq. (1) becomes more pronounced. TX100 can interact with hydrophobic residues and causes the filaments to stay further apart. Thus, the bond breaks at lower forces. Consequently, this further confirms that there is a hydrophobic contribution to the interactions in CB.

In contrast to TX100, magnesium strengthens the bond and keeps it closed even at higher forces as it is a divalent counterion between two negative charges. It lowers the distance to the transition state (compare Fig. 4b blue to gray) and the influence

of the force-dependent term in Eq. (1). Hence, the opening of the bond depends less on the applied force compared to bonds in pure CB. Since CB already includes 2 mM magnesium, we assume that there is an electrostatic contribution to the interactions observed in CB as well.

We have shown that there are hydrophobic and electrostatic contributions to the interactions between IFs and microtubules and we have derived key parameters of these interactions by combining experiments with theoretical modeling. While we cannot exclude a steric contribution to the interaction, e.g., by the Gaussian cloud formed on the surface of the filament core by the intrinsically disordered tail domains of the protein, our measurements show that they are influenced by electrostatic and hydrophobic effects. We thus conclude that the interactions are modulated by electrostatics and hydrophobicity and are directionally independent (see Supplementary Fig. 5 and Supplementary Movie 6). Furthermore, IFs assembled via dialysis are rather smooth[43], so that it is unlikely that their roughness causes interactions.

**Monte-Carlo simulations of dynamic microtubules**. To better understand how these interactions lead to the observed changes in microtubule dynamics, we again applied a modeling approach. We considered a microtubule as a dynamic lattice with GTP (guanosine triphosphate) and GPD (guanosine diphosphate) dimers[20,26] as sketched in Fig. 5a. The lattice consists of 13 protofilaments and has a seam between the first and thirteenth protofilaments. We describe the microtubule dynamics by three reactions: (i) a GTP dimer associates with a rate $r_g$, (ii) a GTP dimer is hydrolyzed with a rate $r_{\text{hy}}$, or (iii) a GDP or GTP dimer dissociates with a rate $r_{\text{dd}}$ or $r_{\text{dt}}$, respectively, depending on the number of neighboring dimers (see Eq. (12) in the Supplementary Information). A snapshot of the simulated microtubule during growth is shown in Fig. 5b. With a Monte-Carlo simulation, we obtained typical simulated kymographs (Fig. 5c). As for the experiments (semi-transparent data in Fig. 5d–f), we determined the growth rate, the catastrophe and the rescue frequency from the simulations (opaque in Fig. 5d–f).

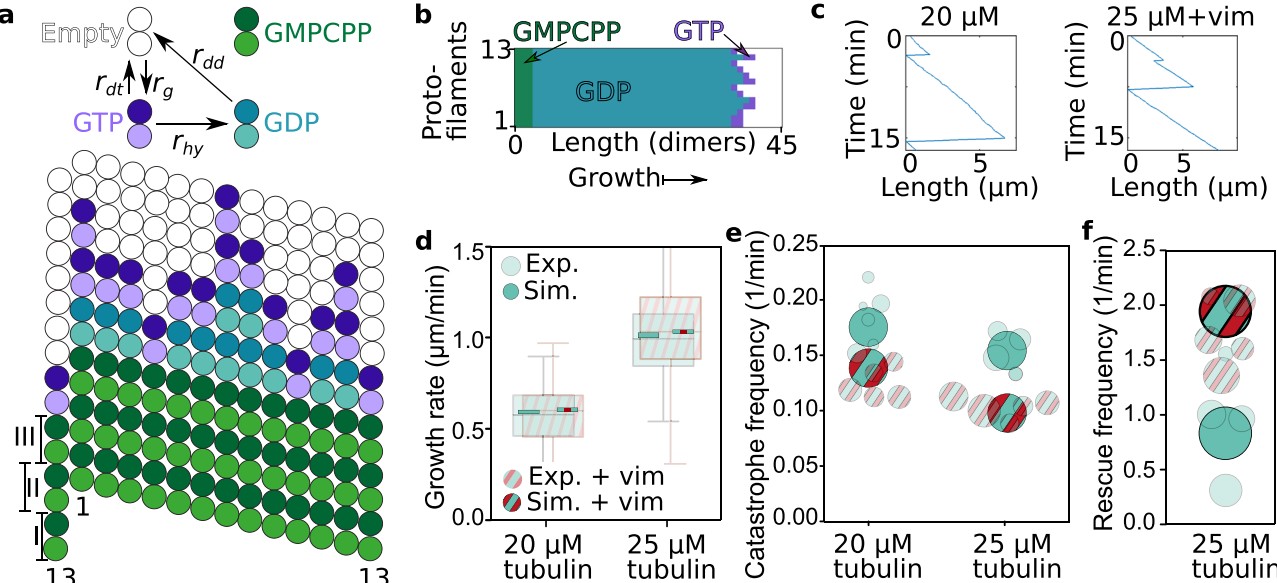

**Fig. 5 A Monte-Carlo simulation shows that transiently binding IFs stabilize dynamic microtubules. a** Illustration of the reaction rates (top) and simulated microtubule lattice with 13 protofilaments and a seam with a longitudinal displacement of 1.5 dimers (bottom). **b** Typical simulated microtubule growing from a GMPCPP (guanylyl-($\alpha,\beta$)-methylene-diphosphonate) seed with dimers either in the GTP (purple) or in the GDP state (cyan). **c** Typical length-time plot (kymograph) of a simulated microtubule in 20 µM free tubulin solution without vimentin tetramers (left) or in 25 µM free tubulin solution with 2.3 µM vimentin tetramers (right). **d–f** We reproduced the experimental data shown in Fig. 1d–g (shown here in a semi-transparent fashion, for a vimentin concentration of 2.3 µM) with our Monte-Carlo simulation (opaque). **d** Addition of vimentin neither changes the experimental nor the simulated microtubule growth rates at 20 or 25 µM. Boxplots include the median as the center line, the 25th and 75th percentiles as box limits, and the entire data range as whiskers. For clarity, the entire data range of the experimental data is not shown here, but is presented in Fig. 1. **e** Addition of vimentin lowers the catastrophe frequency of dynamic microtubules for both tubulin concentrations studied here. **f** In case of 25 µM free tubulin, the rescue rate increases due to the stabilizing effect of the surrounding vimentin IFs. The circle areas scale with the total microtubule depolymerization time as in the representation of the experimental data. All tubulin and vimentin concentrations are input concentrations. Source data are provided as a Source Data file.

To simulate microtubules in the presence of vimentin IFs, we included stochastic binding and unbinding of IFs to the microtubule lattice into the model. The unbinding rate is directly given by the results of the OT experiments. Assuming that binding is diffusion-limited, we calculated the corresponding rate from the OT results using the Smoluchowsky expression (see the Supplementary Information for a detailed description of the model). Based on these rates, we calculated the IF-microtubule binding energy (see Eq. (9) in the Supplementary Information) to be $\Delta G_{\text{IF-MT}} = 2.3 \ k_B T$ as sketched in Fig. 4b. Thus, IF binding stabilizes the binding of tubulin dimers in the microtubule lattice by 2.3 $k_B T$. The additional binding energy can be interpreted as a direct increase of the total binding energy of the respective tubulin dimer or as an increased longitudinal binding energy to the next tubulin dimer. Our experiments do not resolve the precise molecular interaction mechanism, such as cross-linking of neighboring tubulin dimers or structural changes in the tubulin dimers upon binding of a vimentin IF. Likewise, we cannot distinguish whether the vimentin IF is bound to a single tubulin dimer or to multiple dimers. However, our coarse description approach includes all these different scenarios. Specifically, in case of a bond involving multiple dimers, unbinding from these dimers must be cooperative since we do not observe step-wise unbinding in OT experiments. Such cooperativity does not change the total energy required for unbinding. In agreement with our experimental data, the transient binding of IFs leaves the growth rate unaffected. Intriguingly, we observed that IF binding to tubulin dimers in the lattice reduces the catastrophe frequency. The increased binding energy of a dimer also raises the rescue frequency. These results are in striking agreement with our observation in TIRF experiments, while the only additional input to the simulation that includes the surrounding vimentin IFs are the parameters from OT experiments. Thus, stochastic, transient binding of IFs to microtubules as in the OT experiments is sufficient to explain the observed changes in microtubule dynamics in the presence of IFs.

By combining the results from OT and TIRF experiments, we estimated the total binding energy of a tubulin dimer within the lattice at the microtubule tip before catastrophe. From the IF-microtubule bond-breaking events in the OT experiments, including the corresponding simulations, we calculated the IF-microtubule bond energy $\Delta G_{\text{IF-MT}} = 2.3 \ k_B T$ and the unbinding rate $r_{u,\text{eff}}$ of microtubules and vimentin IFs (Fig. 6a). From the TIRF experiments, we determined the catastrophe frequency $f_{\text{cat,IF-MT}}$ of a microtubule bound to a vimentin IF. At the beginning of the catastrophe, a vimentin IF unbinds from the tubulin dimer, so that the energy $\Delta G_{\text{IF-MT}}$ is released. Simultaneously, the dimer depolymerizes from the lattice and the energy $\Delta G_{tb}$ is set free (Fig. 6b). The only additional energy released during microtubule catastrophe in the TIRF experiments compared to the OT experiments is the binding energy to the surrounding tubulin dimers (Fig. 6c). Thus, comparing the rates of IF-microtubule unbinding and microtubule catastrophe during binding to a vimentin IF, as given by Eq. (15) in the Supplementary Information, results in an estimation of the average tubulin-binding energy $\Delta G_{tb}$ between 5.7 $k_B T$ and 7.2 $k_B T$ in the lattice at the tip. These values for $\Delta G_{tb}$ are on the order of magnitude expected from interferometric scattering microscopy and from computational studies, although slightly lower, possibly due to different buffer conditions[20,27,44]. Our combination of experiments provides a way of determining such binding energies and may, from a broader perspective, be generally applied to proteins that bind to microtubules.

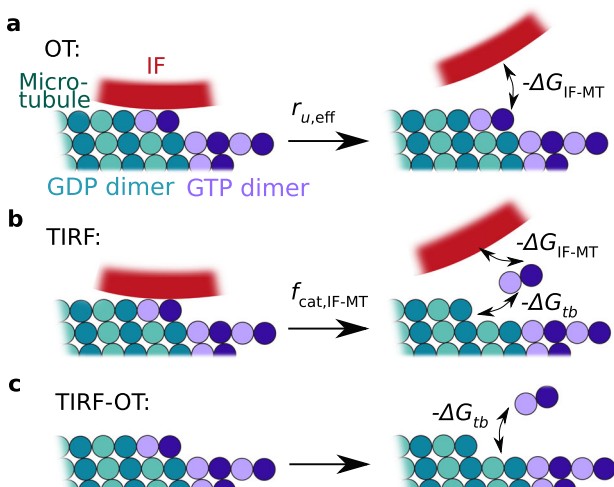

**Fig. 6 Estimation of the binding energy of a single tubulin dimer by combining the OT and TIRF experiments. a** From the OT experiments including the simulations, we determined the unbinding rate of microtubules and vimentin IFs $r_{u,\text{eff}}$ and the released energy $-\Delta G_{\text{IF-MT}}$ during unbinding. **b** In TIRF experiments, a tubulin dimer dissociates from the microtubule and the vimentin IF, so that the total energy $-\Delta G_{tb} - \Delta G_{\text{IF-MT}}$ is released. We calculated the catastrophe frequency of a microtubule $f_{\text{cat,IF-MT}}$ in case a vimentin IF is bound to the microtubule. **c** We estimated the average binding energy $\Delta G_{tb}$ of a tubulin dimer in the microtubule lattice before catastrophe by subtracting the released energies from (**b**), $-\Delta G_{tb} - \Delta G_{\text{IF-MT}}$, and by dividing the unbinding rate $r_{u,\text{eff}}$ of microtubules and vimentin IFs by the catastrophe frequency $f_{\text{cat,IF-MT}}$ of microtubules bound to a vimentin IF as shown by Eq. (15) in the Supplementary Information.

## Discussion

Our study examined the interactions between microtubules and vimentin IFs. We showed that vimentin IFs stabilize microtubules by direct interactions, which is in strong contrast to previous findings[14], where only interactions between microtubules and short IF peptides were considered. Whereas the microtubule growth rate remains unchanged, the stabilization by vimentin IFs leads to a reduction in the catastrophe frequency and increased rescue of depolymerizing microtubules. We pinpoint the source of this stabilizing effect to a stochastic, transient binding of IFs to microtubules by directly measuring the interactions of single filaments. Both hydrophobic and electrostatic effects are involved in bond formation. The presence of cations likely contributes to the attractive interactions between the negatively charged filaments. The buffer in which we conducted the measurements contained potassium and magnesium, two of the most abundant cations in cells[45]. The free magnesium concentration is on the order of a few mM in most mammalian cells[46], similar to our experiments. Magnesium ions have been previously described to cross-link vimentin IFs[47–51], and we showed that they can modulate the IF-microtubule bond strength. Since our magnesium concentrations were close to physiological values, the magnesium-induced IF-microtubule binding we observed may occur in cells as well. Therefore, although molecular motors and cross-linkers contribute to establishing links between IFs and microtubules in cells, our results indicate that more fundamental, direct attractive interactions may also participate in the crosstalk of the two cytoskeletal subsystems in cells.

Gan et al.[18] reported the stabilization of the microtubule network by their co-aligning with vimentin IFs in migrating cells. Complementary to this finding, we showed microtubule stabilization by transient interactions with IFs at the single-filament level without coalignment or bundling of the two filament types. In particular, we found that vimentin IFs and microtubules do not spontaneously coalign. Indeed, according to our estimate based on the measured probability of a vimentin ULF to bind to a tubulin dimer and the number of binding sites per vimentin persistence length, such bundling is highly unlikely (see Supplementary Information): Co-alignment requires interactions at more than one site within one persistence length of a vimentin filament to occur, since thermal fluctuations set the relevant length scale for tight contact between the filaments. This indicates that the cell has to activate additional interaction mechanisms, e.g., via proteins, to induce the coalignment in migrating cells. The rapid, but unfrequent binding of IFs and microtubules we observed suggests that only certain microtubules and IF subunits can bind. Thus, we hypothesize that controlling which subunits can bind (e.g., by posttranslational modifications) may provide another path for the cell to regulate the stabilization of microtubules by IFs. Moreover, our results suggest the possibility that the local vimentin filament concentration, itself controlled by transport of filament fragments along microtubules,[52] may locally tune the dynamic instability of microtubules.

There is growing evidence that a mechanical coupling between the cytoskeletal subsystems is necessary for many cellular functions such as polarization, migration, and mechanical resistance[2,53,54]. In particular, vimentin-deficient cells exhibit a less robust microtubule network orientation[18] and stronger microtubule fluctuations[17], and they show impaired migration, contractility, and resistance to mechanical stress[55–57]. Therefore, future research might help to explore the implications of our findings for cell mechanics and function. Furthermore, our study fosters understanding of the emergent material properties of hybrid networks composed of cytoskeletal filaments and provides a basis for interpreting rheology data, including the dynamic properties of the filaments. Moreover, current efforts in synthetic cell research and materials science may benefit from our findings. Our combination of experiments also offers an alternative approach to estimate the tubulin bond energy within the microtubule lattice, which is a vital parameter to understand microtubule dynamics, mechanics, and function[20–27].

## Methods

**Vimentin purification, labeling, and assembly**. Vimentin C328N with two additional glycines and one additional cysteine at the C-terminus was recombinantly expressed[58–60] and stored at −80 °C in 1 mM EDTA (ethylenediamine tetraacetic acid), 0.1 mM EGTA (ethylene glycol-bis (β-aminoethyl ether) -N,N,N′, N′- tetraacetic acid), 0.01 M MAC (methylamine hydrochloride), 8 M urea, 0.15–0.25 M potassium chloride, and 5 mM TRIS (tris (hydroxymethyl) aminomethane) at pH 7.5[60]. After thawing, we labeled the vimentin monomers with the fluorescent dye ATTO647N (AD 647N-41, AttoTech, Siegen, Germany) and with biotin via malemide (B1267-25MG, Jena Bioscience, Jena, Germany) as described in refs. [61–63]. We mixed labeled and unlabeled vimentin monomers, so that in total 4% of all monomers were fluorescently labeled, a maximum of 20% was biotin labeled and all other monomers were unlabeled[59,63]. We reconstituted vimentin tetramers by first dialyzing the protein against 6 M urea in 50 mM phoshate buffer (PB), pH 7.5, and then in a stepwise manner against 0 M urea (4, 2, 0 M urea) in 2 mM PB, pH 7.5[64], followed by an additional dialysis step against 0 M urea, 2 mM PB, pH 7.5, overnight at 10 °C. To assemble vimentin into filaments, we dialyzed the protein into an assembly buffer, i.e., 100 mM KCl, 2 mM PB, pH 7.5, at 36 °C overnight[60,64].

**Tubulin purification and labeling**. We purified tubulin from fresh bovine brain by a total of three cycles of temperature-dependent assembly and disassembly in Brinkley buffer 80 (BRB80 buffer; 80 mM PIPES (piperazine-N,N′-bis (2-ethanesulfonic acid)), 1 mM EGTA, 1 mM MgCl₂, pH 6.8, plus 1 mM GTP) as described in ref. [65]. After two cycles of polymerization and depolymerization, we obtained microtubule-associated protein (MAP)-free neurotubulin by cation-exchange chromatography (1.16882, EMD SO₃, Merck, Darmstadt, Germany) in 50 mM PIPES, pH 6.8, supplemented with 0.2 mM MgCl₂ and 1 mM EGTA[66]. We prepared fluorescent tubulin (ATTO488- and ATTO565-labeled tubulin; AttoTech AD488-35 and AD565-35, AttoTech) and biotinylated tubulin (NHS-biotin, 21338, Thermo Scientific, Waltham, MA, USA) according to ref. [67]. In brief, microtubules

were polymerized from neurotubulin at 37 °C for 30 min, layered onto cushions of 0.1 M NaHEPES (sodium (4-(2-hydroxyethyl)-1-piperazineethanesulfonic acid), pH 8.6, 1 mM MgCl₂, 1 mM EGTA 60% v/v glycerol, and sedimented by centrifugation at 250,000 × g at 37 °C for 1 h. We resuspended the microtubules in 0.1 M Na-HEPES, pH 8.6, 1 mM MgCl₂, 1 mM EGTA, 40% v/v glycerol, and labeled the protein by adding 1/10 volume of 100 mM NHS-ATTO or NHS-biotin for 10 min at 37 °C. We stopped the labeling reaction by adding 2 volumes of BRB80x2, containing 100 mM potassium glutamate and 40% v/v glycerol. Afterwards, we centrifuged the microtubules through cushions of BRB80 containing 60% v/v glycerol. We resuspended the microtubules in cold BRB80 and performed an additional cycle of polymerization and depolymerization before we snap-froze the tubulin in liquid nitrogen and stored it in liquid nitrogen until use.

**Microtubule seeds for TIRF experiments**. We prepared microtubule seeds at 10 μM tubulin concentration (30% ATTO-565-labeled tubulin and 70% biotinylated tubulin) in BRB80 supplemented with 0.5 mM GMPCPP at 37 °C for 1 h. We incubated the seeds with 1 μM taxol for 30 min at room temperature and then sedimented them by centrifugation at 100,000 × g for 10 min at 37 °C. We discarded the supernatant and carefully resuspended the pellet in warm BRB80 supplemented with 0.5 mM GMPCPP and 1 μM taxol. We either used seeds directly or snap-froze them in liquid nitrogen and stored them in liquid nitrogen until use.

**Sample preparation for OT experiments**. We prepared stabilized microtubules with biotinylated ends for OT by first polymerizing the central part of the microtubules through step-wise increase of the tubulin concentration. Initially, a 3 μM tubulin (5% ATTO-488-labeled) solution in M2B buffer (BRB80 buffer supplemented with 1 mM MgCl₂) in the presence of 1 mM GMPCPP (NU-405L, Jena Bioscience) was prepared at 37 °C to nucleate short microtubule seeds. Next, the total tubulin concentration was increased to 9 μM in order to grow long microtubules. To avoid further microtubule nucleation, we added 1 μM tubulin at a time from a 42 μM stock solution (5% ATTO-488-labeled) and waited for 15 min between the successive steps. To grow biotinylated ends, we added a mix of 90% biotinylated and 10% ATTO-565-labeled tubulin in steps of 0.5 μM from a 42 μM stock solution up to a total tubulin concentration of 15 μM. We centrifuged the polymerized microtubules for 10 min at 13,000 × g to remove any non-polymerized tubulin and short microtubules. We discarded the supernatant and carefully resuspended the pellet in 800 μL M2B-taxol (M2B buffer supplemented with 10 μM taxol (T7402, Merck)). By keeping the central part of the microtubules biotin-free (see the color code in Fig. 2a: biotin-free microtubule in cyan and biotin-labeled microtubule ends in green), we ensured that any streptavidin molecules detaching from the beads could not affect the interaction measurements by cross-linking the filaments.

For measurements in the microfluidic chip by OT, we prepared four solutions for the four different microfluidic channels as sketched in Fig. 2a: (I) We diluted streptavidin-coated beads with an average diameter of 4.5 μm (PC-S-4.0, Kisker, Steinfurt, Germany) 1:83 with the vimentin assembly buffer. (II) We diluted the vimentin IFs 1:667 with the vimentin assembly buffer. (III) We diluted the resuspended microtubules 1:333 with CB. (IV) We combined suitable buffer conditions for microtubules and for vimentin IFs, respectively, to a combination buffer (CB) containing 1 mM EGTA (03777, Merck), 2 mM magnesium chloride, 25 mM PIPES (9156.4, Carl Roth, Karlsruhe, Germany), 60 mM potassium chloride (6781.3, Carl Roth) and 2 mM sodium phosphate (T879.1 and 4984.2, Carl Roth) at pH 7.5. We included an oxygen scavenging system consisting of 1.2 mg/mL glucose (G7528, Merck), 0.04 mg/mL glucose oxidase (G6125-10KU, Merck), 0.008 mg/mL catalase (C9322-1G, Merck), and 20 mM DTT (dithiothreitol, 6908.2, Carl Roth). Additional 0.01 mM taxol (T1912-1MG, Merck) was added to stabilize the microtubules. For measurements with TX100, we added 0.1% (w/v) Triton-X 100 (TX100; 3051.3, Carl Roth) and in case of measurements with a total magnesium concentration of 20 mM, we added 18 mM MgCl₂. We filtered the solutions through a cellulose acetate membrane filter with a pore size of 0.2 μm (7699822, Th. Geyer, Renningen, Germany).

**OT experiments**. We performed the OT experiments using a commercial setup (C-Trap, LUMICKS, Amsterdam, The Netherlands) equipped with quadruple optical tweezers, a microfluidic chip, and a confocal microscope. We used the Bluelake software (version b11, LUMICKS) to conduct the experiments. The beads, microtubules, measuring buffer and IFs were flushed into four inlets of the microfluidic chip as sketched in Fig. 2a. For each measurement, four beads were captured and then calibrated in the buffer channel using the thermal noise spectrum. One bead pair (beads 1 and 2) was moved to the vimentin IF channel and incubated there until a filament bound to the beads (Fig. 2aII.). Meanwhile, the other bead pair (beads 3 and 4) was kept in the measuring buffer channel, so that no filaments adhered to those beads. To capture a microtubule (Fig. 2aIII.), beads 3 and 4 were moved to the microtubule channel, while beads 1 and 2 stayed in the measuring buffer channel. Once a microtubule was bound to beads 3 and 4 and an IF to beads 1 and 2, the bead pair with the IF was horizontally turned by 90° (Supplementary Fig. 3a) and moved up in the z-direction by 4.9 μm. The bead pair holding the IF was moved in the x-y plane so that the central part of the IF was

positioned above the center of the microtubule (Fig. 2aIV. and Supplementary Fig. 3a). To bring the IF and microtubule into contact, the IF was moved down in the z-direction until the microtubule was pushed into focus or slightly out of focus. In a portion of the experiments, we turned the microtubule by 45° and visually controlled the angle by fluorescence microscopy.

The IF was moved perpendicularly to the microtubule in the y-direction in the x-y plane at 0.55 μm/s, while we measured the forces in the x- and y-direction on bead 1. For a horizontal movement, we moved the IF perpendicularly to the microtubule in the x-direction in the x-y plane at 0.55 μm/s. Simultaneously, we recorded confocal images to see whether an interaction occurred. In case no interaction occurred after two movements in the x-y plane, the IF was moved down in the z-direction by 0.4 μm and the movement in the x-y plane was repeated. The experiment ended when the microtubule broke off the bead, or the IF or microtubule broke. In case of the vertical movement in a perpendicular configuration, we measured 57 pairs of microtubules and vimentin IFs in CB, 38 pairs with TX100, and 36 pairs with additional magnesium chloride. In total, we moved the IFs 744 times vertically and perpendicularly to the microtubules in CB, 704 times in CB with TX100, and 542 times in CB with additional magnesium chloride. In the case of the 45°-configuration we studied 49 pairs of microtubules and vimentin IFs and completed 467 movements. In the case of horizontal movement of the IF in perpendicular configuration with respect to the microtubule, we studied 43 filament pairs and completed 504 movements.

**OT data analysis**. The OT data were processed with self-written Matlab (Math-Works, Natick, MA, USA) scripts. In case of vimentin IF movement in the y-direction and a perpendicular orientation to the microtubule, we analyzed the component of the force $F_{1y}$ acting on bead 1 in the y-direction for each filament pair, since the forces in the x-direction were balanced by the IF, as sketched in Supplementary Fig. 3b. From the raw force data, we manually selected the force data containing an interaction. Due to the interactions of the energy potentials of the different traps, some data sets exhibited a linear offset, which we subtracted from the data. From the interaction-free force data, we determined the experimental error by calculating the standard deviation in the force of the first 20 data points. We defined an interaction as soon as the force $F_{1y}$ as shown in Fig. 2d deviated by more than $5\sigma_F$ from the mean of the first 20 data points, where $\sigma_F$ is the standard deviation of the force without interactions in each data set. Typically, the force increased as shown in Fig. 2d until the interaction ended with a fast force decrease as marked by $\Delta F_{1y}$. We did not take breaking forces below 0.5 pN into account because they may be caused by force fluctuations. Since the force detection of trap 1 is the most accurate one in the setup, we analyzed the force on bead 1 only. To determine the total breaking force $F_B$, we multiplied the force $F_{1y}$ acting on bead 1 in y-direction with a correction factor $c_F$ that is based on the geometry of the experiment. $c_F$ depends on the distance $d_{MT}$ between beads 1 and 2 and the distance $d_{IF-MT}$ from bead 1 to the contact point of the IF and the microtubule as sketched in Supplementary Fig. 3c:

$$c_F = \frac{d_{MT}}{d_{IF-MT}} \ . \tag{3}$$

For the total force $F_C$ acting on the IF-microtubule bond, we get

$$F_C = c_F F_{1y} \ .$$

Thus, when an IF-microtubule bond breaks, the total force difference $F_B$ is

$$F_B = c_F \Delta F_{1y} \ . \tag{4}$$

In case of a 45° angle between the microtubule and the vimentin IF and in case of IF movement in the x-direction, the force data were analyzed in the same way as described above with the following differences:

For the case where the microtubule was turned by 45°, we calculated the geometric factor in two different ways, depending on the geometry at the moment of bond breakage: (i) If the bond breaks at a higher y-position than bead 2 (see Supplementary Fig. 3d), we assume that the total force acting on the IF-microtubule bond is measured by the complete force $F_1$ acting on bead 1. Thus, $c_F = 1$, and the breaking force is

$$F_B = \Delta F_1 \ .$$

(ii) If the bond breaks at a lower y-position than bead 2 (see Supplementary Fig. 3e), the geometric factor is

$$c_F = 1 + \frac{\tan \alpha_2}{\tan \alpha_1} \ .$$

We analyzed $F_{1y}$ again and the breaking force was calculated with Eq. (4).

In case the IF was moved perpendicularly to the microtubule in the x-direction, we analyzed the force $F_{1x}$ acting on bead 1 in the x-direction. We did not need to correct for the geometry of the experiment, thus, $c_F = 1$:

$$F_B = \Delta F_{1x} \ .$$

**Preparation of passivated cover glasses for TIRF experiments**. We cleaned cover glasses (26 × 76 mm², no. 1, Thermo Scientific) by successive chemical treatments: (i) We incubated the cover glasses for 30 min in acetone and then (ii)

for 15 min in ethanol (96% denatured, 84836.360, VWR, Radnor, PA, USA), (iii) rinsed them with ultrapure water, (iv) left them for 2 h in Hellmanex III (2% (v/v) in water (Hellma Analytics, Müllheim, Germany), and (v) rinsed them with ultrapure water. Subsequently, we dried the cover glasses using nitrogen gas flow and incubated them for three days in a 1 g/L solution of 1:10 silane-PEG-biotin (PJK-1919, Creative PEG Works, Chapel Hill, NC, USA) and silane-PEG (30 kDa, PSB-2014, Creative PEG Works) in 96% ethanol and 0.02% v/v hydrochloric acid, with gentle agitation at room temperature. We subsequently washed the cover glasses in ethanol and ultrapure water, dried them with nitrogen gas, and stored them at 4 °C for a maximum of 4 weeks.

**TIRF experiments**. We used an inverted microscope (IX71, Olympus, Hamburg, Germany) in TIRF mode equipped with a 488-nm laser (06-MLD, 240 mW, COBOLT, Solna, Sweden), a 561-nm laser (06-DPL, 100 mW, COBOLT), and an oil immersion TIRF objective (NA = 1.45, 150×, Olympus). We observed microtubule dynamics by taking an image every 5 s for 15–45 min using the cellSens Dimensions software (version 1.18, Olympus) and a digital CMOS camera (ORCA-Flash4.0, Hamamatsu Photonics, Hamamatsu, Japan).

For TIRF experiments, we built flow chambers from passivated cover glasses and a double-sided tape (70 μm height, 0000P70PC3003, LiMA, Couzeix, France). We flushed 50 μg/mL neutravidin (A-2666, Invitrogen, Carlsbad, CA, USA) in BRB80 into the chamber and incubated for 30 s. To remove free neutravidin, we washed with BRB80. Afterwards, we flushed microtubule seeds diluted 300 × in BRB80 into the chamber and incubated for 1 min before we removed free-floating seeds by washing with BRB80 supplemented with 1% BSA (bovine serum albumin). Then, a mix containing 0.5 mg/mL or 0.8 mg/mL (corresponding to 2.34 or 3.74 μM) vimentin tetramers (4% ATTO-565-labeled), 20 or 25 μM tubulin dimers (20% ATTO488-labeled), 0.65% BSA, 0.09% methyl cellulose, 2 mM phosphate buffer, 2 mM MgCl₂, 25 mM PIPES, 1 mM EGTA, 60 mM KCl, 20 mM DTT, 1.2 mg/mL glucose, 8 μg/mL catalase, and 40 μg/mL glucose oxidase, pH 7.5, was perfused into the chamber. To avoid evaporation and convective flow, we closed the chamber with vacuum grease and placed it on the stage of the TIRF microscope that was kept at 37 °C. We used the cellSens Dimensions software (version 1.18, Olympus).

From the TIRF movies, kymographs were created using the reslice function of ImageJ (ImageJ V, version 2.0.0-rc-69/1.52p). From the kymographs, microtubule growth and depolymerization velocities, and catastrophe and rescue frequencies were estimated. We calculated the catastrophe frequency for each experiment as

$$f_{\text{cat,exp}} = \frac{\text{total number of catastrophe events}}{\text{total microtubule growth time}}$$

and the rescue frequency as

$$f_{\text{resc}} = \frac{\text{total number of rescue events}}{\text{total microtubule depolymerization time}}$$

The total growth time was 800–2000 min per condition, and the total depolymerization time 50–70 min per condition. The growth rate plot in Fig. 1d contains between 106 and 163 measurements per condition.

**Determination of vimentin filament length distributions**. To measure the lengths of vimentin filaments (see Supplementary Fig. 1), we prepared five 1.5 mL reaction tubes with 15 μL of a mix of 2.3 μM vimentin in CB including all additions as used for the TIRF experiments, such as methyl cellulose, GTP, and oxygen scavenger (see the previous section for the exact composition of the buffer). We then incubated the mix at 37 °C for 5, 10, 20, 30, or 45 min. The filament assembly was stopped by adding 25 volumes of buffer to the tubes. Five microliters of each diluted mix was then pipetted on a cover glass and a second cover glass was placed on top. Images were taken with an inverted microscope (IX81, Olympus) using the cellSens Dimensions software (version 1.18, Olympus), a 60× oil-immersion PlanApoN objective (Olympus), and an ORCA-Flash 4.0 camera (Hamamatsu Photonics). The filament lengths were determined using the semi-automated JFilament 2D plugin (Lehigh University, Bethlehem, PA, USA, version 1.02) for ImageJ (version 2.0.0-rc-69/1.52p).

**Modeling**. A detailed description of the modeling approaches is provided in the Supplementary Information.

In brief, breaking of microtubule-IF interactions was modeled as a force-dependent two-state model and distributions of breaking forces were determined both by stochastic simulations and by numerically solving for the first passage time distribution. The time-dependent force accounts for the pulling speed and the entropic elasticity of the filament. The unknown parameters were varied systematically to find regions in parameter space consistent with the OT experiments for different buffer conditions and different measurement geometries.

The simulation of dynamic microtubules was adapted from refs. 20, 26 by additionally considering the binding energy of a microtubule to a vimentin IF obtained from the OT experiments. We thus simulated dynamic microtubules embedded in a vimentin IF network.

The binding energy of a microtubule dimer in the lattice at the tip of a microtubule was calculated from an energy balance comparing the unbinding rate of microtubules and vimentin IFs, and the catastrophe frequency with and without vimentin.

**Reporting summary**. Further information on research design is available in the Nature Research Reporting Summary linked to this article.

## Data availability
Data supporting the findings of this manuscript are available from the corresponding authors upon reasonable request. A reporting summary for this article is available as a Supplementary Information file. Source data are provided with this paper.

## Code availability
The source codes of the numerical simulations and analysis are available at https://data.goettingen-research-online.de/dataverse/schaedel_lorenz_2021.

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

## Acknowledgements

We thank Thomas Wenninger, Helge Schmidt, and Sarah Adio for their support towards the construction of the TIRF setup. We are grateful to Manuel Théry, Laurent Blanchoin, Jérémie Gaillard, and Susanne Bauch for providing the purified tubulin and vimentin. We thank Susanne Bauch, Susanne Hengst, and Ulrike Schulz for technical support.

## Author contributions

S. Köster conceived and supervised the project. S. Köster and L.S. designed the experiments. L.S. and C.L. performed all experiments and analyzed the data. A.V.S. helped in performing the quadruple optical trap experiments. C.L. and S. Klumpp designed and performed the numerical simulations. All authors contributed to writing the manuscript.

## Funding

This work was financially supported by the European Research Council (ERC, Grant No. CoG 724932, to S. Köster), the European Molecular Biology Organization (Long Term Fellowship No. 1164-2018, to L.S.), and the Studienstiftung des deutschen Volkes e.V. (fellowship to C.L.). This research was conducted within the Max Planck School Matter to Life (to S. Köster and S. Klumpp) supported by the German Federal Ministry of Education and Research (BMBF) in collaboration with the Max Planck Society. The work further received financial support via an Excellence Fellowship of the International Max Planck Research School for Physics of Biological and Complex Systems (IMPRS PBCS, fellowship to A.V.S.). Open Access funding enabled and organized by Projekt DEAL. We acknowledge support by the Open Access Publication Funds of Göttingen University.

## Competing interests

The authors declare no competing interests.
