## [Peer Review File · Nature Communications]

Reviewer #1 (Remarks to the Author):

I commend the authors for the detailed responses and extra work in response to the referee comments. I do think this is an excellent paper that explores a physically interesting interaction between microtubules and vimentin intermediate filaments. The paper is well written, experiments and modeling are interesting and well executed and the results appear to be sound.

One minor comment is that the 1d transition model, while totally fine here, may not apply to other filament pairs that may have have intrinsically geometry dependent interactions (due to protein structure). The authors might want to comment.

The fact that the studied interactions are transient and do not lead to bundling (or any observable stable associations) coupled with the existence of other stronger protein-mediated interactions in cells makes the implications for this interaction unlikely *in vivo*. I think, therefore that the authors may be well served by focusing on the implications for such interactions in synthetic protein based materials (hybrid networks of cytoskeletal filaments) , rheology and fundamental measurements of polymerization energy. The emphasis on the significance *in vivo* both in the introduction and conclusion is somewhat counter-productive.

Overall, I think it is publishable in Nature Communications if we only consider the caliber of the experimental and modeling work, but perhaps less so if *in-vivo* significance is deemed essential.

Reviewer #2 (Remarks to the Author):

Schaedel et al use *in vitro* reconstitution assays, optical trap assays and *in silico* simulations and modeling to investigate the effect of vimentin IF to dynamic microtubules *in vitro*. Using *in vitro* assays of microtubule dynamics, they first show that vimentin IF stabilize microtubules by reducing the catastrophe frequency while increasing the rescue frequency. The microtubule growth rate was remained unaffected. Using an optical trap approach, Schaedel et al measure the forces exerted during orthogonal interactions between vimentin filaments and microtubules and how they are affected by alteration of hydrophobic and electrostatic interactions by addition of Triton X100 detergent and magnesium chloride, respectively. Lastly the authors use results from the optical trap and microtubule dynamics assays to derive the binding energy of tubulin dimers within the microtubule lattice.

As pointed out in previous reviews, the results of this work is only significant if placed in the context of cell biological work published by Gan et al in Cell Systems (2016). That paper showed that (1) vimentin filaments co-align with microtubules in living cells and enhance the stability of the latter, (2) in the presence of vimentin, microtubules are more resistant to the microtubule depolymerizing drug nocodazole indicating that vimentin enhances the microtubule stability, and (3) after nocodazole wash out, microtubules grow along vimentin filaments suggesting that microtubule plus ends are captured by and grow along vimentin filaments.

The current work does not provide a significant conceptual advance over the previous observations, which provide critical *in vivo* corollary and some validation or credance to the finding of this

manuscript, which lacks compelling evidence for a direct vimentin filament-microtubule interaction - as also raised by the other two reviewers (and more strongly by Reviewer 3).

There is weak evidence for a direct interaction. In Figure 1, vimentin overlap with microtubules is nearly non-existent and any overlap appears to be random. In the absence of any data showing either vimentin filaments co-aligning with microtubule lattices or microtubule plus ends docking on and moving along vimentin filaments, it cannot be ascertained that the effects of vimentin filaments on microtubules is due to direct interactions.

The Gan et al work in 2016 predicts that microtubules grow along vimentin filaments, and that is something that can be easily observed with the in vitro system (Figure 1) using sparsely immobilized vimentin filaments.

The dot blot assay that the authors provided in their rebuttal does not make a convincing case and it's unclear what exactly demonstrates.

In search of experiments that can provide some evidence of direct interaction, the authors may try to see if they can decorate polymerized microtubules with soluble fluorescent vimentin. It is possible that unpolymerized vimentin may bind microtubules and this can be also assayed using microtubule pelleting assays with soluble vimentin. It is also worth performing this experiment with microtubule lattices bound to GMPCPP, GTP gamma S and GDP in case there are preferences for the nucleotide bound to microtubules. The same can be tried with polymerized vimentin filaments - although Figure 1 does not indicate that this is likely to work.

Statistical analyses, p and n values are missing from Figure 1, which raises concerns about the rigor of this work.

Regretfully, even with some in vitro data supporting a direct interaction between vimentin and microtubules, this reviewer remains skeptical of the overall significance of this work given the previous work published by Gan et al. Clearly, this is a recurrent disagreement between this reviewer and the authors, and it will be something that the editor(s) will have to resolve as we have reached an impasse.

Reviewer #3 (Remarks to the Author):

Positive aspects of the revised manuscript, now intended for Nature Communications:

The authors have improved the manuscript to the best of their ability. The work is performed and presented with care and rigor. The observations and their implications are interesting, and if the conclusions that the authors make from these data are correct, the work has clear cell biological significance. I agree that the manuscript should be published somewhere and will be a useful

foundation for future work; Nature Communications seems like an appropriate destination.

Having said this, I still have some concern about the validity of the work.

1. Even after the additional work and arguments provided by the authors, it still seems that an equally reasonable way to explain the transient "binding" events detected in the trap assay is to suggest that they are steric in nature: if you drag one polymer against another, the one being dragged is likely to occasionally snag on the other, especially if one can "fray" or have pieces that could potentially stick up (e.g., as might be expected of the globular heads of IFs).

2. The observation that concerns me most is that the authors do not observe bundling or other association between IFs and MTs in fluorescent imaging. I recognize that the authors explain the lack of bundling between IFs and MTs by saying that the interactions between the polymers are "transient," and they now provide in the supplement an explanation based on the persistence length of IFs. This is an important discussion, and it may indeed provide the necessary explanation. However, if it is correct, it then raises the question of how (given that there are so few interactions between IFs and MTs) the IFs can have the observed effects on MT dynamics.

3. I agree that the reported effects on catastrophe and rescue provide strong evidence that IFs are indeed binding to MTs. However, given the other concerns above, I remain worried that there is an artifactual explanation for this observation, such as unrecognized environmental changes that occur upon addition of the IF protein. I say this not out of spite, but out of painful experiences in my own lab: protein function (especially tubulin polymerization) can be very sensitive to minor and unintentional differences in salt and/or pH caused by addition of a test protein.

Thus, I have two additional minor requests before publication:

1. That the authors add to the main text a brief discussion providing their explanation for how the polymers can bind each other without being seen to co-associate; this discussion should reference their analysis in the Supplementary Information.

2. That the authors more explicitly address how the number and duration of interactions between IFs and MTs is consistent with the observed effects on MT dynamics. I assume that this is present in their modeling work, but I had difficulty finding an explicit discussion of this important point.

Point-by-point response to the comments by the reviewers

Reviewer #1 (Remarks to the Author):

I commend the authors for the detailed responses and extra work in response to the referee comments. I do think this is an excellent paper that explores a physically interesting interaction between microtubules and vimentin intermediate filaments. The paper is well written, experiments and modeling are interesting and well executed and the results appear to be sound. One minor comment is that the 1d transition model, while totally fine here, may not apply to other filament pairs that may have have intrinsically geometry dependent interactions (due to protein structure). The authors might want to comment. We thank the reviewer for this comment. Indeed, it is correct that we do not observe any direction/geometry dependency in our experiments and thus opted for the 1d transition model to describe our data. If, instead, we studied a system with such a dependency, probably a more complicated model would be necessary, or a 1d model would be valid for one pulling direction only. We now added a corresponding comment on page 7.

The fact that the studied interactions are transient and do not lead to bundling (or any observable stable associations) coupled with the existence of other stronger protein-mediated interactions in cells makes the implications for this interaction unlikely in vivo. I think, therefore that the authors may be well served by focusing on the implications for such interactions in synthetic protein based materials (hybrid networks of cytoskeletal filaments) , rheology and fundamental measurements of polymerization energy. The emphasis on the significance in vivo both in the introduction and conclusion is somewhat counter-productive.

We thank the reviewer for this suggestion. In response, we have added a stronger reference to the importance of studying in vitro systems to the introduction on page 3 as well as to the conclusions on page 13 and have taken out some text referring to the cellular relevance.

Overall, I think it is publishable in Nature Communications if we only consider the caliber of the experimental and modeling work, but perhaps less so if in-vivo significance is deemed essential.

Reviewer #2 (Remarks to the Author):

Schaedel et al use in vitro reconstitution assays, optical trap assays and in silico simulations and modeling to investigate the effect of vimentin IF to dynamic microtubules in vitro. Using in vitro assays of microtubule dynamics, they first show that vimentin IF stabilize microtubules by reducing the catastrophe frequency while increasing the rescue frequency. The microtubule growth rate was remained unaffected. Using an optical trap approach, Schaedel et al measure the forces exerted during orthogonal interactions between vimentin filaments and microtubules and how they are affected by alteration of hydrophobic and electrostatic interactions by addition of Triton X100 detergent and magnesium chloride, respectively. Lastly the authors use results from the optical trap and microtubule dynamics assays to derive the binding energy of tubulin dimers within the microtubule lattice.

As pointed out in previous reviews, the results of this work is only significant if placed in the context of cell biological work published by Gan et al in Cell Systems (2016). That paper showed that (1) vimentin filaments co-align with microtubules in living cells and enhance the stability of the latter, (2) in the presence of vimentin, microtubules are more resistant to the microtubule depolymerizing drug nocodazole indicating that vimentin enhances the microtubule stability, and (3) after nocodazole wash out, microtubules grow along vimentin filaments suggesting that microtubule plus ends are captured by and grow along vimentin filaments.

The current work does not provide a significant conceptual advance over the previous observations, which provide critical in vivo corollary and some validation or credance to the finding of this manuscript, which lacks compelling evidence for a direct vimentin filament-microtubule interaction - as also raised by the other two reviewers (and more strongly by Reviewer 3).

We respectfully disagree with the Reviewer. As pointed out by Reviewer 1 and mentioned in our manuscript on pages 3 and 13, our findings are highly relevant in important fields of science such as fundamental biophysics (hybrid networks of (cytoskeletal) biopolymers, rheology, polymerization kinetics), materials science (synthetic protein-based materials) or synthetic cell research. We still think that the Gan et al. work is a good motivation and starting point for our current work. The results of Gan et al. and ours are complementary, as Gan et al. address stabilization at the network level, while we show stabilization of individual dynamic filaments.

We also disagree with the reviewer's statements (here and below) about the lack of strong evidence for a direct interaction between microtubules and vimentin filaments. Our optical trap experiments, including movies which visualize the process clearly, provide strong evidence for a direct interaction and lead to a quantitative assessment of the corresponding forces.

There is weak evidence for a direct interaction. In Figure 1, vimentin overlap with microtubules is nearly non-existent and any overlap appears to be random. In the absence of any data showing either vimentin filaments co-aligning with microtubule lattices or microtubule plus ends docking on and moving along vimentin filaments, it cannot be ascertained that the effects of vimentin filaments on microtubules is due to direct interactions.

The Gan et al work in 2016 predicts that microtubules grow along vimentin filaments, and that is something that can be easily observed with the in vitro system (Figure 1) using sparsely immobilized vimentin filaments.

The reviewer's comments about the evidence for a direct interaction appear to be based exclusively on our TIRF experiments and the lack of co-alignment or bundling of the filaments in these experiments. However, in our opinion, clear evidence for a direct interaction is given by our quantitative interaction experiments using optical traps, from which we derive forces and rates. We use these values for an estimate of more than one bond to occur simultaneously, within one persistence length of a vimentin filament. Our estimate shows that we do, indeed, not expect bundling, due to the transient nature of the rare (but strong!) interactions.

Here, our work and the Gan et al. publication address different aspects. While we study a general interaction phenomenon of two filament types, the living, migrating cells in the Gan et al. paper represent a special and highly relevant biological situation.

The dot blot assay that the authors provided in their rebuttal does not make a convincing case and it's unclear what exactly demonstrates.

We fully agree with the reviewer that the dot plot assay is no convincing demonstration of the interaction between microtubules and vimentin intermediate filaments; in general, such co-sedimentation assays are not suitable to detect transient interactions. In response to an earlier reviewer report, we still conducted that assay and the result is as expected. Thus, our optical trapping assay is much more sensitive to such transient interactions and it is furthermore a quantitative approach providing numerical values for forces and rates.

In search of experiments that can provide some evidence of direct interaction, the authors may try to see if they can decorate polymerized microtubules with soluble fluorescent vimentin. It is possible that unpolymerized vimentin may bind microtubules and this can be also assayed using microtubule pelleting assays with soluble vimentin. It is also worth performing this experiment with microtubule lattices bound to GMPCPP, GTP gamma S and GDP in case there are preferences for the nucleotide bound to microtubules. The same can be tried with polymerized vimentin filaments - although Figure 1 does not indicate that this is likely to work.

We would like to stress again that we have a sensitive, novel method at hand (optical trapping assay) to quantitatively measure transient interactions, supported by clear visualization (see supplementary movies). In this assay, we measure the interactions between filaments (not subunits). Thus, it remains unclear to us, which additional information is expected from the suggested experiments.

Statistical analyses, p and n values are missing from Figure 1, which raises concerns about the rigor of this work.

We have added the numbers of samples N and events n , as well as the total recording times to the caption of Fig. 1.

In Fig. 1d,e, we present data with a large standard deviation, thus comparatively small differences in the mean are not of interest. We have added a corresponding comment to the manuscript on page 4.

For the data in Fig. 1f,g a comparison as suggested by the Reviewer is not trivial as each circle corresponds to many data points and pooling all data would mean to lose information. We thus prefer the current representation of the data.

Regretfully, even with some in vitro data supporting a direct interaction between vimentin and microtubules, this reviewer remains skeptical of the overall significance of this work given the previous work published by Gan et al. Clearly, this is a recurrent disagreement between this reviewer and the authors, and it will be something that the editor(s) will have to resolve as we have reached an impasse.

We indeed still disagree with the Reviewer on this point. Our results are new observations and have not been shown yet. Specifically, that vimentin affects the dynamic instability of microtubules has not been described before and was also not expected based on previous work. Previous literature (e.g. Gan et al. 2016) shows that in migrating epithelial cells, microtubules and vimentin IFs bundle. However, in non-migrating cells, microtubules and vimentin IFs do not form bundles and their interaction is unknown. By contrast to the work by Gan et al., we study microtubules and vimentin IFs on the single filament level and observe a stabilization even without bundling and we propose an interaction mechanism. Our results are relevant for in vitro studies of filaments, which are commonly performed by the biophysics community, and for possible cellular mechanisms tuning the interaction of microtubules and vimentin IFs.

Reviewer #3 (Remarks to the Author):

Positive aspects of the revised manuscript, now intended for Nature Communications: The authors have improved the manuscript to the best of their ability. The work is performed and presented with care and rigor. The observations and their implications are interesting, and if the conclusions that the authors make from these data are correct, the work has clear cell biological significance. I agree that the manuscript should be published somewhere and will be a useful foundation for future work; Nature Communications seems like an appropriate destination.

Having said this, I still have some concern about the validity of the work.

1. Even after the additional work and arguments provided by the authors, it still seems that an equally reasonable way to explain the transient "binding" events detected in the trap assay is to suggest that they are steric in nature: if you drag one polymer against another, the one being dragged is likely to occasionally snag on the other, especially if one can "fray" or have pieces that could potentially stick up (e.g., as might be expected of the globular heads of IFs).

Indeed, the surface of intermediate filaments is decorated by the intrinsically disordered tail domains of the protein (possibly with some involvement of the heads), forming a "Gaussian cloud" around the filament core (we have now added this information to page 9). In principle, this could add to steric interactions, it is, however, unlikely that they result in a completely geometry- and direction-independent effect.

2. The observation that concerns me most is that the authors do not observe bundling or other association between IFs and MTs in fluorescent imaging. I recognize that the authors explain the lack of bundling between IFs and MTs by saying that the interactions between the polymers are “transient,” and they now provide in the supplement an explanation based on the persistence length of IFs. This is an important discussion, and it may indeed provide the necessary explanation. However, if it is correct, it then raises the question of how (given that there are so few interactions between IFs and MTs) the IFs can have the observed effects on MT dynamics.

In response to this important point, we have added a short comment on page 12.

3. I agree that the reported effects on catastrophe and rescue provide strong evidence that IFs are indeed binding to MTs. However, given the other concerns above, I remain worried that there is an artifactual explanation for this observation, such as unrecognized environmental changes that occur upon addition of the IF protein. I say this not out of spite, but out of painful experiences in my own lab: protein function (especially tubulin polymerization) can be very sensitive to minor and unintentional differences in salt and/or pH caused by addition of a test protein.

This is an important comment by the reviewer. We have taken all possible care during our experiments: the combination buffer contains 25 mM Pipes and is thus quite highly concentrated so as to buffer the solution well. We do not add any additional salt when adding the vimentin. Beyond this, we cannot think of a suitable control experiment to compare the situation with and without vimentin present.

Furthermore, our main evidence for the direct interactions between microtubules and vimentin filaments stems from the optical trap experiments, where we can directly measure and quantify the interaction forces.

Thus, I have two additional minor requests before publication:

1. That the authors add to the main text a brief discussion providing their explanation for how the polymers can bind each other without being seen to co-associate; this discussion should reference their analysis in the Supplementary Information. We thank the Reviewer for this suggestion and have added a corresponding comment to page 12.

2. That the authors more explicitly address how the number and duration of interactions between IFs and MTs is consistent with the observed effects on MT dynamics. I assume that this is present in their modeling work, but I had difficulty finding an explicit discussion of this important point.

The number and duration of interactions enter the model of microtubule dynamics, which indeed shows the consistency with the observed effect of vimentin filaments on microtubule dynamics, through binding and unbinding rates of vimentin filaments to microtubules. We have rephrased the corresponding text on page 10 to now state explicitly how these rates are determined from the optical trap data and refer to the Supporting Information for a detailed description.